# Tribological Properties of Polyimide Composites Modified with Diamondoid Metal–Organic Frameworks

**DOI:** 10.3390/polym16060806

**Published:** 2024-03-14

**Authors:** Zihui Yu, Xianqiang Pei, Qianyao Pei, Yan Wang, Zhancheng Zhang, Yaoming Zhang, Qihua Wang, Tingmei Wang

**Affiliations:** 1State Key Laboratory of Solid Lubrication, Lanzhou Institute of Chemical Physics, Chinese Academy of Sciences, Lanzhou 73000, China; m13936725058@163.com (Z.Y.); peiqianyao@licp.cas.cn (Q.P.); wangyan3@licp.cas.cn (Y.W.); zhangzhancheng@licp.cas.cn (Z.Z.); yaomingzhang@licp.cas.cn (Y.Z.); wangqh@licp.cas.cn (Q.W.); wangtm@licp.cas.cn (T.W.); 2Center of Materials Science and Optoelectronics Engineering, University of Chinese Academy of Sciences, Beijing 100049, China; 3Yantai Zhongke Research Institute of Advanced Materials and Green Chemical Engineering, Yantai 264000, China

**Keywords:** metal-organic framework, polyimide, tribological properties, transfer films, composites

## Abstract

In this work, diamondoid metal-organic frameworks (MOFs) were efficiently prepared by sonochemical synthesis and grown on polyimide (PI), aiming to improve the anti-wear performance of the PI matrix. By introducing MOFs into the PI matrix, the free movement of PI molecular chains were restricted, and its hardness and elastic modulus were improved. It was found that the wear rate of the 3 wt.% MOFs/PI composites was reduced by 72.6% compared to pure PI at a load of 4 N after tribological testing by using a ball-on-disk tribometer. This can be attributed to the excellent load-bearing and shear resistance of the fourfold-interpenetrated diamondoid networks, in which the transition metal elements can favor the formation of transfer films. It is worth noting that the 3 wt.% MOFs/PI composites still exhibited great tribological properties under high loads or high speeds. The findings of the present study indicate that diamondoid metal-organic frameworks can be used as efficient modifiers to enhance the tribological properties of PI.

## 1. Introduction

Polyimide (PI) is a high-performance polymer matrix that has been developed since 1950s, and it has excellent low-temperature resistance [1], high-temperature resistance [2], radiation resistance [3], and excellent self-lubricating properties [4]; as a result, it is widely used in the aerospace industry [5], triboelectric nanogenerators [6], and radiation-shielding materials [7]. However, the coefficient of friction and wear rate of pure PI are relatively high, which limits its application in extreme environments [8]. Fillers can be used to improve the tribological properties of pure PI, such as carbon fibers [9], graphene [10], PTFE [11], zirconia [12]. But the interfacial adhesion between fillers and polymer matrix is usually weak due to their chemical inertness, which leads to the easy detachment of fillers from the polyimide matrix during friction. In this case, three-body abrasive wear might be induced, and the composites’ anti-wear performance are reduced. 

Metal-organic frameworks (MOFs) are crystalline materials with a periodic network structure formed by the interconnection of inorganic metal centers (metal ions or metal clusters) and bridging organic ligands [13]. MOFs have been found to show many advantages, such as high porosity, tunable architecture, good thermal stability, high specific surface area, and so on, due to which they have been paid much attention in academic societies [14]. In addition to applications in the fields of adsorption and separation [15], catalysis [16], sensing [17], and energy storage and conversion [18], MOFs have also been verified to be effective in the area of tribology. Wu et al. [19] have used MOFs@DDP as additives for base oils, and they revealed its good anti-friction and anti-wear effects, which were ascribed to the ball-bearing effect of MOFs in the friction process. Sui et al. [20] have applied ZIF-8 nanoparticles as Si_3_N_4_ ceramic water lubrication additives and found that it mainly improved the lubrication properties of silicon nitride ceramics through filling and film-forming mechanisms. In addition, the presence of organic ligands in MOFs can potentially improve their properties compared to traditional inorganic nanomaterials [21] in such a way that they have helped enhance the interfacial bonding between fillers and the matrix. This has been proved to be beneficial for the mechanical and tribological properties of the composites [22]. Additionally, Lin et al. [23] have found that transition metal elements are conducive to the formation of carbon-based transfer films, which provides a theoretical basis for MOFs to improve the tribological properties of polymers. The above advantages promote the application of MOFs in the field of tribology and make MOFs a new type of promising tribological modifier.

Compared to ordinary MOFs, diamondoid MOFs consist of a multiple interpenetration structure, which can further enhance the interface between MOFs and polymer molecules. However, the application of diamondoid MOFs as fillers is limited due to their relatively lower production rate as well as their non-optimized synthesis technology [13]. To overcome the above problems, the sonochemical synthesis method was developed to significantly shorten the crystallization time and reduce the particle size of MOFs [24]. With the incorporation of ultra-small MOF particles into the polymer matrix, the formation of defects can be effectively reduced around the particles. Bearing this in mind, diamondoid metal-organic frameworks were sonochemically synthesized and added to polyimide with the aim of improving its mechanical and tribological properties. In order to guide the application of diamondoid MOFs as tribological modifiers in the polymer matrix, we used the ball-on-disk tribometer to evaluate the tribological properties of MOFs/PI composites and discussed their friction and wear mechanisms.

## 2. Materials and Methods

### 2.1. Materials

4,4′-Oxydianiline (ODA) was purchased from Shanghai Macklin Biochemical Technology Co., Ltd. (Shanghai, China). Zinc nitrate hexahydrate, ethanol absolute, and ether were provided by Sinopharm Chemical Reagent Co., Ltd. (Shanghai, China). Urocanic acid was purchased from Sigma-Aldrich (Shanghai, China) Trading Co., Ltd. Quinoline, N-methylpyrrolidone (NMP), and 4,4′-(hexafluoroisopropylidene)diphthalic anhydride (6FDA) were produced by Aladdin Reagent (Shanghai, China) Co., Ltd. Deionized water was obtained from a laboratory ultrapure water machine.

### 2.2. Preparation of MOFs

MOFs were prepared by using a modified sonochemical synthesis method in reference to existing schemes [25]. The detailed operation steps were performed as follows: first, we dissolved 5.95 g of zinc nitrate hexahydrate in 100 mL of water, then weighed 2.76 g of urocanic acid, and dispersed it in 100 mL (5:1) (ethanol absolute: water) solution to form a homogeneous dispersion. Under the condition of magnetic stirring and temperature of 0~4 °C, the zinc nitrate hexahydrate solution was added dropwise to the urocanic acid dispersion at a speed of 4 mL/min through a peristaltic pump, and the ultrasonic probe dispersion equipment was turned on at the same time. After the reaction continued for one hour, the obtained substances were successively washed with hot water, ethanol absolute, and ether. Finally, the white powdered MOFs were dried at room temperature. 

### 2.3. Preparation of PI and MOFs/PI Composites

A two-step method was used to synthesize the PI or MOFs/PI composites [26]. Initially, ODA was added to 47 mL of NMP solution, and the corresponding proportions of MOFs (0 wt.%, 1 wt.%, 3 wt.%, 5 wt.%, 7 wt.%) were added when ODA was completely dissolved. The MOFs were evenly dispersed in the solution with mechanical stirring for 10 min. We slowly added the corresponding proportion of 6FDA under the conditions of nitrogen protection and mechanical stirring. After the reaction continued for 24 h, quinoline was added to the obtained solution and mechanical stirring continued for one hour to obtain a poly(amic acid) (PAA) solution. Subsequently, the PAA solution was coated on the GCr15 bearing steel matrix; baked in a vacuum drying box at 50 °C for three hours; and then heated up to 90 °C for two hours, 160 °C for one hour, and 180 °C for one hour. The coatings were naturally cooled down to room temperature to obtain the PI or MOFs/PI composites.

### 2.4. Test Methods

#### 2.4.1. Mechanical Tests

According to the national standard GB/T 25898-2010, the hardness and elastic modulus of the MOFs/PI composites were tested by using a nanoindentation instrument (STeP E400, Anton Paar, Graz, Austria). We repeated this at least three times to ensure the reliability of the data. The maximum load of the nanoindentation test was 3 mN, and the dwell time was 10 s. The hardness was defined as follows [27]: $H=\frac{P_{max}(h_{max})}{A(h_{c})}$. Here, H is the hardness of the composites, *P_max_* is the maximum applied load, h_max_ is the maximum penetration depth, A is the contact area, and h_c_ is the contact depth. The elastic modulus can be calculated from the reduced modulus and the indenter modulus by the following formula [27]:$\;\frac{1}{E_{r}}=\frac{\left(1-v^{2}\right)}{E}+\frac{\left(1-v_{i}^{2}\right)}{E_{i}}$, $E_{r}=\frac{\surd \pi }{2\beta }\frac{S}{\surd A}$. Here, E_r_ is the reduction modulus, S is the contact stiffness, β is a constant related to the geometry of the indenter, and v is the Poisson’s ratio for the composites.

#### 2.4.2. Tribological Properties of PI and MOFs/PI Composites

Referring to ASTM G99-2017, the tribological properties of the composites were tested on a ball-on-disk tribometer (MFT5000, Rtec, San Jose, CA, USA). Tribological tests were conducted on composites with different contents of MOFs (0 wt.%, 1 wt.%, 3 wt.%, 5 wt.%, 7 wt.%), under different loads (4 N, 8 N, 12 N, 20 N, 25 N, 30 N), and at velocities ranging from 0.1 m/s to 0.25 m/s. The tribological properties test was carried out at room temperature, and the counterpart was a GCr15 bearing steel ball with a diameter of 5 mm, which circularly rotated in a radius of 10 mm. The duration of each friction test was set to 100 min, and the test was repeated at least 3 times to ensure the reliability of the data. The coefficient of friction was the average value when friction entered the stabilization phase. The sliding velocity was calculated according to the following formula: v=2π·R·n/60. Where ν is the linear velocity during friction, R is the radius of rotation, and n is the rotational speed of the sample (r/min). The composites’ volumetric wear rate (W_s_) was calculated using the following formula:$\;W_{S}=\frac{\Delta V}{F_{N\cdot L}}\;$(mm^3^/Nm), ΔV=2πRS. S was calculated by observing the morphology of the worn surfaces by using a 3D Optical Profilers, where R is the rotation radius, F_N_ is the load, and L is the sliding distance.

### 2.5. Analysis and Characterization Techniques

The prepared MOF crystal structure was analyzed by using X-ray diffraction (Ultima IV, Rigaku, Akishima, Japan) measurements in the 2θ range of 5°~60°; the scanning speed was set to 5°/min, and the test temperature was 25 °C. Fourier transform infrared spectroscopy (TENSOR 27, Burker, Bremen, Germany) was used to analyze the structure of the composites and MOFs. The morphology of MOF particles, MOFs/PI composites, and their counter steel-ball-worn surfaces were characterized via SEM (JSM−7610F, JEOL, Tokyo, Japan). The samples were observed at 5 kV (counter steel ball surface) and 15 kV (composites and MOF nanoparticles), respectively. In order to improve the electrical conductivity of the composites, a layer of gold film was applied to the surface of the materials by using a magnetron sputtering device before testing.

## 3. Results and Discussion

### 3.1. Morphology and Structure of the Materials

Figure 1a shows an SEM image of the MOFs prepared by sonochemical synthesis. It can be seen that the particle size of the MOFs prepared by using this method is between 700 nm and 1 μm, with a regular octahedron structure. By correlating the simulated and measured XRD diffraction patterns of the prepared MOFs (Figure 1b), it is clear that the MOFs with the designed structure were successfully synthesized. Different modes of FTIR were used to characterize the main functional groups of the materials. As can be seen from Figure 1c, peaks at 1700 cm^−1^ and 3440 cm^−1^ were associated with the tensile vibration peaks and telescopic vibration peaks of C=O bonds and O−H bonds, which corresponded to the urocanic acid ligand in the MOFs [25]. The asymmetrical and symmetrical stretching vibration peaks of C=O appeared in PI and its composites at 1772 cm^−1^ and 1721 cm^−1^, and the stretching vibration peaks of C−N−C of the imide rings appeared at 1360 cm^−1^. There were no obvious characteristic absorption peaks at 3500 cm^−1^–3100 cm^−1^, which proved that the thermal imidization process of polyimide was complete (Figure 1d). It is worth noting that the absorption peak of the carboxyl group in the 3 wt.% MOFs/PI composites completely disappeared, indicating the involvement of the MOFs’ carboxylic acid ligand in the synthesis reaction of polyimide. This was beneficial for the enhancement of interfacial bonding between the filler and polyimide matrix [28].

### 3.2. Mechanical Properties of MOFs/PI Composites

The load-depth curves of the nanoindentation test and the micromechanical properties of the composites are shown in Figure 2a,b. When the same load was applied, the indentation depth of the composites was first decreased and then increased with increasing MOF content (Figure 2a). It is clear that the hardness of the composites was higher than that of pure PI when a small amount of the MOFs was added. Apparently, this improvement can be ascribed to the in situ growth of MOFs in the PI matrix, which limited the free movement of the PI molecular chains and led to effective stress transmission and strengthened mechanical properties due to strong interfacial bonding [22,29]. However, the incorporation of a high content of fillers into the PI matrix induced more stress concentrations and microcracks under the action of external forces, thus reducing the mechanical properties of the composites [22]. As shown in Figure 2b, the hardness and elastic modulus of the MOFs/PI composites were first increased and then decreased with the increase in filler content. When the MOF content was 5 wt.%, the elastic modulus and hardness of the MOFs/PI composites were increased to 5.238 GPa and 0.474 GPa, respectively, which accounted for improvements of 7.22% and 11.44% compared with pure PI. The improved micromechanical properties contributed to the enhancement of the wear resistance of the composites, as is discussed in the following.

### 3.3. Tribological Properties of MOFs/PI Composites

Figure 3a shows the evolution of the friction coefficient of the MOFs/PI composites with sliding time. As can be seen from the figure, the friction process of all the materials can be divided into two stages: the running-in phase and the steady phase. The running-in time of pure PI lasted for 2000 s, after which the friction coefficient leveled off at about 0.35 with small fluctuations in the friction curve. When the MOFs were added to the polyimide matrix, the running-in time was shortened and the fluctuation in the friction curve was abated, except for with 1 wt.% MOFs/PI composite. This variation in friction behavior was related to the fact that the organic-inorganic interface structure of the MOFs/PI was conducive to alleviating asperity deformation and enhancing the friction stability of the composites [30]. It is worthy of noting that it took only 700 s for the composites with 3 wt.% MOFs to reach the steady state, which can be attributed to the accelerated formation of stable transfer films induced by the MOFs [23]. The variation in the friction coefficient and wear rate of the MOFs/PI composites with content of MOFs is shown in Figure 3b. With the increase in MOF content, the composites’ wear rate was firstly decreased and then increased, which concaved at contents of 3 wt.%–5 wt.%. A minimum wear rate of 2.13 × 10^−5^ mm^3^/Nm was registered for the 3 wt.% MOFs/PI composite, which was 72.6% lower than that of the pure PI. This can be attributed to the excellent load-bearing and shear resistance of the diamondoid networks, in which transition metal elements can favor the formation of transfer films [23]. In contrast to the obviously increased wear resistance, the friction coefficient of PI was increased slightly. This was related to the mutually constraining role of the MOFs in the PI matrix. On the one hand, the addition of MOFs improved the shear resistance of the material surface, which tended to increase the friction coefficient; on the other hand, the favored formation of transfer films lowered the friction by preventing the direct contact between composite surface and counter ball. Consequently, a slightly higher friction torque was induced in the present study, which resulted in a slight increase in the coefficient of friction [28].

In order to explore the friction and wear mechanisms of the MOFs/PI composites with different contents of MOFs, the worn surface morphology of the composites and the corresponding transfer films formed on the counter steel ball surfaces were observed by using a scanning electron microscope (as shown in Figure 4). As can be seen from Figure 4a, there were large microcracks on the worn surface of pure PI, which resulted from fatigue fracture under repeated interfacial shearing. This led to the generation of granular abrasive debris that adhered to the surface of the counter steel ball to form sparsely distributed sheet-like transfer films (Figure 4b), which peeled off under the action of shear forces, causing the coefficient of friction to greatly fluctuate during the sliding process. The above morphology changed significantly with the addition of the MOFs. When the fillers content was 3 wt.%, the area of the microcrack zone on the worn surface was significantly smaller and accompanied by the traces of furrows, and the MOF particles distributed in the PI matrix can be seen on the worn surface (Figure 4c). In this case, the MOFs can increase not only the ability of the composite surface to resist shear forces, but also to bear the load during friction process and inhibit the damage of the counter steel ball to the composite surface. At the same time, relatively homogeneous and continuous transfer films were formed on the counter steel ball surface (Figure 4d). It is worthing pointing out that the chelate reaction between the transition metal elements in MOFs and the counter steel balls can promote the formation of carbon-based transfer films and increase the interfacial adhesion between the transfer film and counter steel ball surface. This is deemed to be the main reason for the significantly enhanced wear resistance of the MOFs/PI composites [31]. When the content of MOFs was increased to 7 wt.%, parallel furrows were accompanied with scratches inclined to the sliding direction on the composites’ worn surface (Figure 4e), which were related to the three-body abrasive wear caused by the detached MOFs at the sliding interface. The occurrence of three-body abrasive wear led to the destruction of the transfer films on the counter steel ball surface (Figure 4f) and the deterioration of their anti-wear performance. This was responsible for the increased wear rate of the composites at higher MOF contents.

The tribological properties of the MOFs/PI composites were further investigated by exploring the friction and wear of 3 wt.% MOFs/PI composites under different loads and velocities. Figure 5a shows the friction curves of the 3 wt.% MOFs/PI composites at a velocity of 0.1 m/s under different loads. It is clear that all the friction curves rapidly came to a plateau when the friction process entered a stabilized phase. It should be noted that the 3 wt.% MOFs/PI composites exhibited two stabilized friction phases under 30 N. This should be attributed to the fact that the transfer films formed in the first phase were destroyed after a certain time of sliding under such a high load, after which they were formed again rapidly and the second steady state was reached. The increase in the friction coefficient in the following steady phase was associated with the increased contact area at the extended sliding time. In the present study, 30 N can be considered as the critical value for the MOFs/PI composites, above which the detachment of transfer films overwhelmed their formation and stabilized sliding could not be maintained anymore. As for the load dependency of the composite’s running-in behavior, it was found to be decreased with the increase in load, which may be related to the fact that the high load promoted the decomposition of MOFs, thereby accelerating the formation of transfer films. It has to be pointed out that the composite’s wear rate was more sensitive to load than the friction coefficient, as can be seen from Figure 5b. For the former, it was decreased in a linear manner from 2.13 × 10^−5^ mm^3^/Nm to 1.1 × 10^−5^ mm^3^/Nm as the load increased from 4 N to 20 N. Further increases in load toward the critical value of 30 N worsened the composite’s wear resistance, but the wear rate was still lower than that observed under 4 N, indicating the excellent anti-wear performance of the MOFs/PI composites, even under higher loads.

Figure 6 presents the worn surface morphology of the 3 wt.% MOFs/PI composite under different loads. Compared with the worn surface morphology at 4 N (Figure 4c), when the load was increased to 20 N, there appeared obvious signs of fracture on the worn surface of the composite (Figure 6a), which was due to the fact that the PI molecular chains were stretched until they broke under the high interfacial shear force. The wear debris generated in this process was repeatedly squeezed and deformed, and it adhered to the counter steel ball surface to finally form dense and homogeneous transfer films (Figure 6b). This effectively reduced the direct contact between the composite’s surface and the counter steel ball, which was responsible for the improved wear resistance of the studied composite. As the load was further increased to 30 N, the worn surfaces of the composite was plastically deformed so severely that they stacked on top of each other (Figure 6c). Correspondingly, the transferred materials on the counterpart surface were extruded out of the contact zone, resulting in a decrease in the area covered by the transfer films on the counter steel ball surface (Figure 6d). This is consistent with the increased wear rate of the studied composite under such a high load (Figure 5b).

Under a fixed load of 12 N, the tribological properties of the 3 wt.% MOFs/PI composite at different sliding speeds were investigated. As can be seen from Figure 6a, the composite’s running-in time was prolonged with the increase in velocity. At the highest sliding speed of 0.25 m/s in the present study, it was registered to be ca. 1000 s, which is nearly 5 times that observed at speeds lower than 0.2 m/s. This was presumably due to the instability of the transfer films formed at the beginning of sliding, which were constantly peeled off under the action of the rapidly repeated shear of the counter body. In this case, a longer time was needed to form stable transfer films as the sliding speed was increased. It is worth mentioning that the curve fluctuated greatly due to the continuous peeling off and formation of the transfer films. The friction coefficient of the 3 wt.% MOFs/PI composite was hardly dependent on velocity, but the wear rate was first decreased and then increased with velocity (Figure 7b). When the velocity was 0.2 m/s, the composite’s wear rate reached the minimum of 1.25 × 10^−5^ mm^3^/Nm.

Figure 8 shows the worn surface morphology of the MOFs/PI composites and corresponding counter steel balls at different velocities. It can be seen from the figure that when the sliding speed was 0.1 m/s, obvious furrows and main cracks covered the worn surface of the composite (Figure 8a), which indicated that abrasive wear and fatigue wear were the main wear mechanisms. At the same time, the discontinuous transfer film with traces of breaks can be observed on the counter steel ball surface (Figure 8b), and the larger area of exposed steel damaged the composite’s surface and led to a high wear rate. With the velocity increased to 0.2 m/s, the composite’s worn surface was characterized with an abated formation of furrows and main cracks (Figure 8c), and relatively continuous transfer films were observed on the counter steel surface (Figure 8d), which was responsible for the minimum wear rate, as was discussed beforehand. However, when the velocity was increased to 0.25 m/s, the worn surface of the composite was full of furrow marks and main cracks, which may have been due to the fact that under the action of rapid shear force, the composite material was peeled off and sheared into particles in the friction interface (Figure 8e). This led to the occurrence of three-body abrasive wear with the formation of narrow furrows on the worn surface of the composite, against which the continuous transfer films were broken to some extent. In this case, the exposed area of the counter steel was increased (Figure 8f), and the composite’s anti-wear performance was consequently worsened.

## 4. Conclusions

Diamondoid MOFs were successfully prepared by using sonochemical synthesis and used as modifiers to improve the mechanical and tribological properties of polyimide. The main conclusions are as follows:(1)An appropriate amount of diamondoid MOFs can improve the hardness and elastic modulus of the PI matrix due to the restricted molecular movement and enhanced interfacial bonding between MOFs and PI molecules. The modifying role of MOFs became impaired at excessive incorporations of them, and a content of 3 wt.% was recommended according to the present study.(2)The MOF particles helped increase the anti-shear performance of the PI surface and can bear the load during friction, during which relatively homogeneous and continuous transfer films were formed on the counterpart surface. As a combinational consequence, the wear rate of PI was decreased effectively, while the friction coefficient was increased slightly.(3)At the optimal content of 3 wt.%, the MOFs/PI composite still exhibited excellent tribological properties under high load or high velocity. This can mainly be attributed to the formation of uniform and continuous transfer films on the counterpart surface, which were promoted by the decomposition of MOFs.

## Figures and Tables

**Figure 1 polymers-16-00806-f001:**
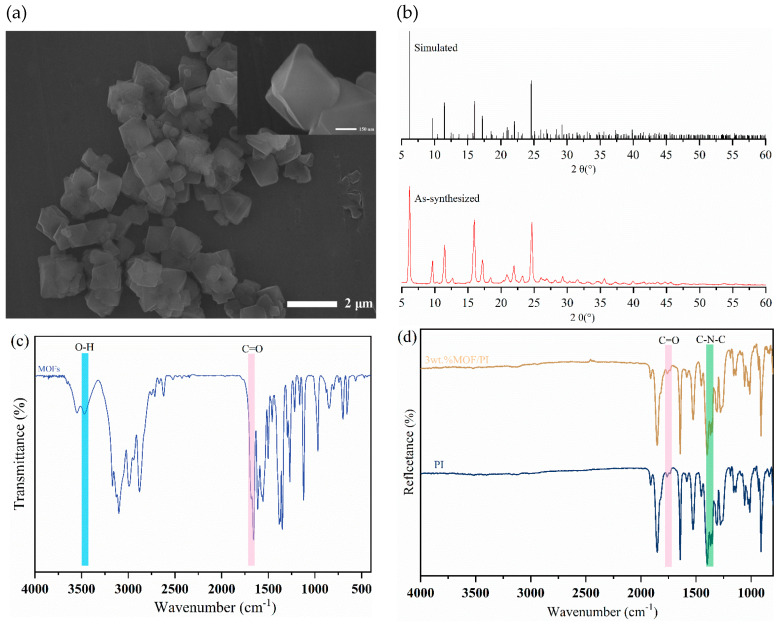
(**a**) SEM image of MOFs. (**b**) X-ray diffraction patterns of MOFs. (**c**) Transmission infrared spectra of MOFs. (**d**) Attenuated total reflection infrared spectra of the composites.

**Figure 2 polymers-16-00806-f002:**
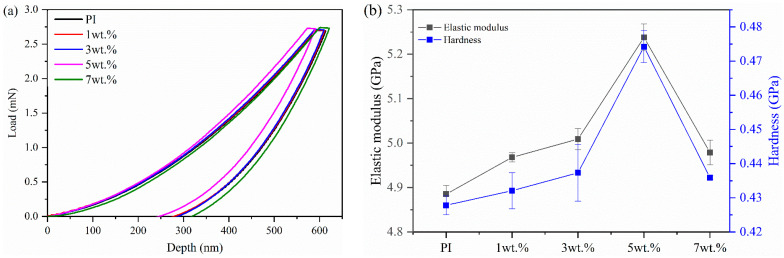
(**a**) Load-depth curves of nanoindentation test. (**b**) Variation curves of hardness and elastic modulus of the composites with MOF content.

**Figure 3 polymers-16-00806-f003:**
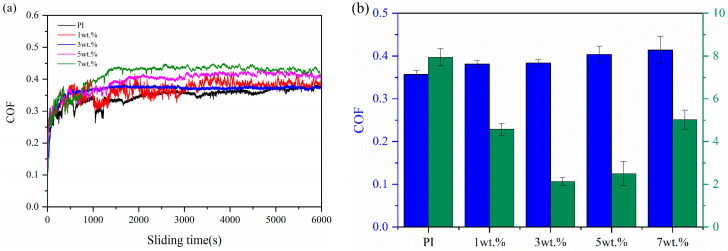
(**a**) Variation curves of friction coefficient with sliding time. (**b**) MOFs/PI composites coefficient of friction and wear rate with content of MOFs.

**Figure 4 polymers-16-00806-f004:**
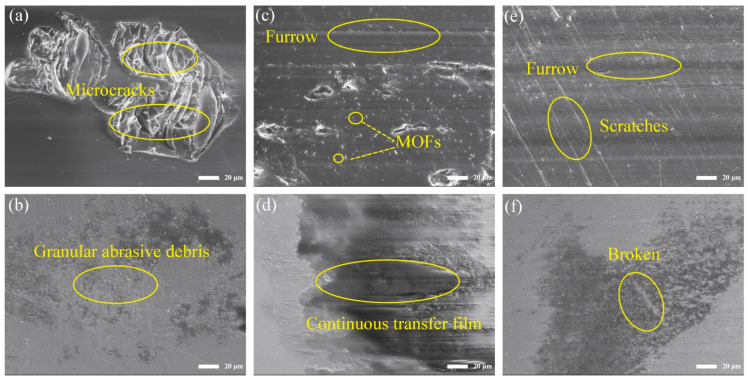
SEM images of MOFs/PI composites worn surface (top) and counter steel ball surface (bottom): (**a**,**b**) PI, (**c**,**d**) 3 wt.% MOFs/PI, (**e**,**f**) 7 wt.% MOFs/PI.

**Figure 5 polymers-16-00806-f005:**
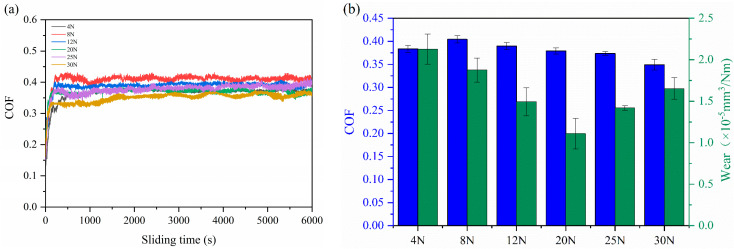
(**a**) Variation curves of friction coefficient with sliding time, (**b**) coefficient of friction, and wear rate with load of 3 wt.% MOFs/PI composites.

**Figure 6 polymers-16-00806-f006:**
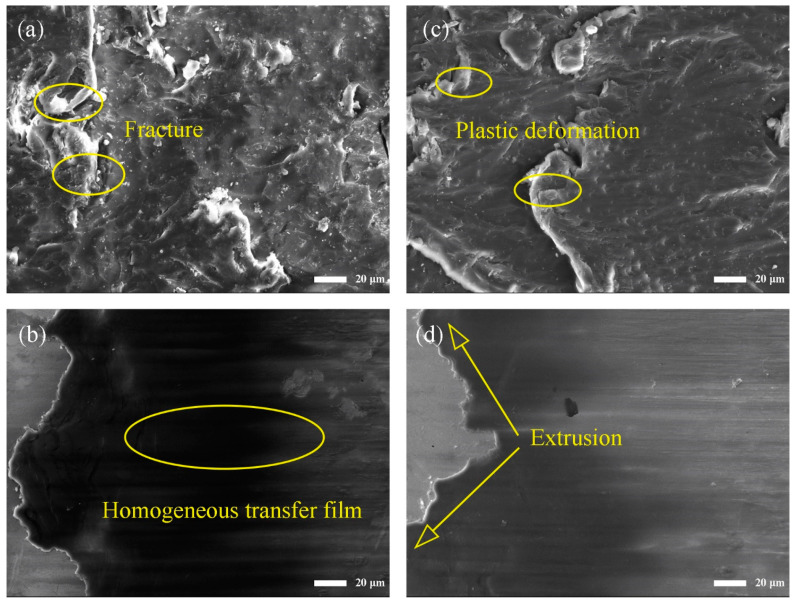
SEM images of 3 wt.%MOFs/PI composites (top) and counter steel surface (bottom) with different loads. (**a**,**b**) 20 N, (**c**,**d**) 30 N.

**Figure 7 polymers-16-00806-f007:**
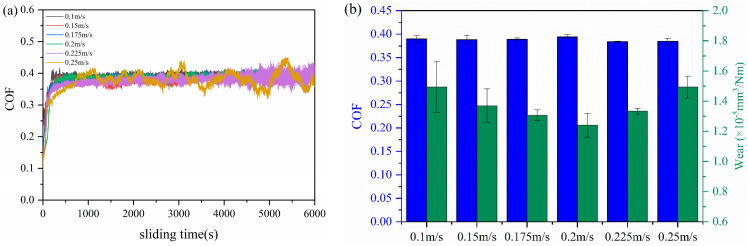
(**a**) Variation curve of friction coefficient with sliding time, (**b**) coefficient of friction and wear rate of 3 wt.% MOFs/PI composites at different velocities.

**Figure 8 polymers-16-00806-f008:**
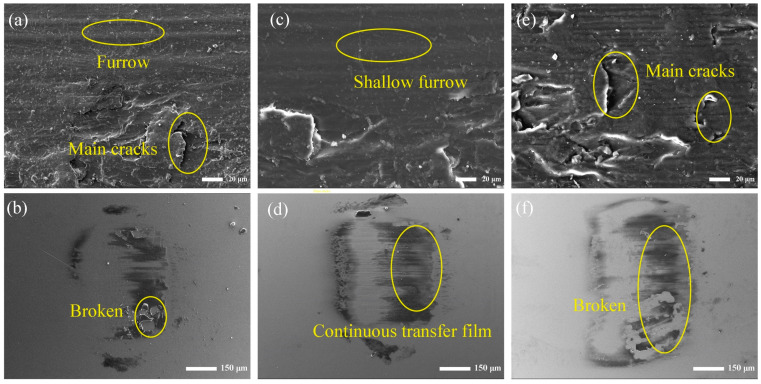
SEM images of the 3 wt.% MOFs/PI composites (top) and counter steel ball surface (bottom) with different velocities. (**a**,**b**) 0.1 m/s, (**c**,**d**) 0.2 m/s (**e**,**f**) 0.25 m/s.

## Data Availability

Data are contained within the article.

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
