# Peer review of "Tribological Properties of Polyimide Composites Modified with Diamondoid Metal–Organic Frameworks"

_polymers, 2024, doi:10.3390/polym16060806_

Round 1

Reviewer 1 Report

Comments and Suggestions for Authors

1)    The methodology of the work in brief is to be added in the ABSTRACT instead elaborating on the result only. 

2)    The objective and novelty of the research is mentioned at the end of Introduction section.

3)    Figures 1a, 4 and 6, SEM image, the details are missing except scale bar. Why details are masked or removed? Please add.

4)    Figure 2a “Load-Depth curves of nanoindentation test”, the number of trials used to record the data is to be mentioned.

5)    It is mentioned in the explanation of Figure 5a that “Further increase of load to the critical value of 30 N worsened the composite’s wear resistance, but the wear rate was still lower than that under 4 N indicating excellent anti-wear performance of the MOFs/PI composites even under higher loads”. Give technical reason for the statement.

Please format the Reference section. In some of the paper titles (Ref. no. 3, 10, 12, 14, 15 and 17), the 1st letter of each word is of upper-case letter whereas in other references it is different. Also, journal name is fully written with Upper case (capital) letters in some places. Please format all the references

Comments on the Quality of English Language

1)    The methodology of the work in brief is to be added in the ABSTRACT instead elaborating on the result only. 

2)    The objective and novelty of the research is mentioned at the end of Introduction section.

3)    Figures 1a, 4 and 6, SEM image, the details are missing except scale bar. Why details are masked or removed? Please add.

4)    Figure 2a “Load-Depth curves of nanoindentation test”, the number of trials used to record the data is to be mentioned.

5)    It is mentioned in the explanation of Figure 5a that “Further increase of load to the critical value of 30 N worsened the composite’s wear resistance, but the wear rate was still lower than that under 4 N indicating excellent anti-wear performance of the MOFs/PI composites even under higher loads”. Give technical reason for the statement.

Please format the Reference section. In some of the paper titles (Ref. no. 3, 10, 12, 14, 15 and 17), the 1st letter of each word is of upper-case letter whereas in other references it is different. Also, journal name is fully written with Upper case (capital) letters in some places. Please format all the references

Author Response

Dear Reviewer,

Thank you for your carefully review of our manuscript (polymers-2921735). We sincerely appreciate your very helpful comments and suggestions. The paper has been modified accordingly and all detailed changes in the paper are listed below point by point. We hope that you are satisfied with our responses to the comments. Thanks again for your efforts in processing our manuscript, and look forward to your reply soon!

With best regards,

Yours sincerely,

Xianqiang Pei

Lanzhou Institute of Chemical Physics

No.18 Tianshui Middle Rd., Lanzhou, 73000, China

Comments 1 : The methodology of the work in brief is to be added in the ABSTRACT instead elaborating on the result only. 

Response: Thank you for pointing this out. We agree with this comment. Therefore, we have introduced the method of tribological testing in the Abstract. [It was found that the wear rate of the 3wt.% MOFs/PI composites was reduced by 72.6% compared to pure PI at a load of 4N after tribological testing by using a ball-on-disk tribometer.]

Comments 2 : The objective and novelty of the research is mentioned at the end of Introduction section.

Response: Thank you for your suggestion. We have added a detailed description at the end of Introduction section.[In order to guide the application of diamondoid MOFs as tribological modifier in polymer matrix, we used the ball-on-disk tribometer to evaluate the tribological properties of MOFs/PI composites, and discussed its friction and wear mechanisms.]

Comments 3 :Figures 1a, 4 and 6, SEM image, the details are missing except scale bar. Why details are masked or removed? Please add.

Response: We originally modified the SEM image to make it look aesthetically pleasing and easy to see, without paying attention to the importance of other parameters. We have added other specific parameters to the 2.5 Analysis and characterization techniques section.

Comments 4 : Figure 2a “Load-Depth curves of nanoindentation test”, the number of trials used to record the data is to be mentioned.

Response: We think your suggestion is very necessary, so we added the number of trials in the 2.4.1 Mechanical tests section to make the experiments more convincing. [Repeat at least three times to ensure the reliability of the data.]

Comments 5 : It is mentioned in the explanation of Figure 5a that “Further increase of load to the critical value of 30 N worsened the composite’s wear resistance, but the wear rate was still lower than that under 4 N indicating excellent anti-wear performance of the MOFs/PI composites even under higher loads”. Give technical reason for the statement.

Response: Combined with the SEM images and the structure of MOFs, we speculate that higher loads are conducive to the decomposition of MOFs and promote the formation of excellent transfer films in counterpart. Besides, at lower loads, MOFs will wear the materials as hard particles, which is not conducive to the formation of transfer films. So it is inferred that MOFs/PI composites will exhibit excellent anti-wear under higher loads.

Reviewer 2 Report

Comments and Suggestions for Authors

The authors have mentioned the properties of polyimide but have not mentioned the exact industrial applications. So please add a paragraph mentioning the applications of polyimides.

Everywhere, the authors have used the word tribiological properties. I recommend the use of specific terms like wear rate and friction coefficient.

Author Response

Dear Reviewer,

Thank you for your carefully review of our manuscript (polymers-2921735). We sincerely appreciate your very helpful comments and suggestions. The paper has been modified accordingly and all detailed changes in the paper are listed below point by point. We hope that you are satisfied with our responses to the comments. Thanks again for your efforts in processing our manuscript, and look forward to your reply soon!

With best regards,

Yours sincerely,

Xianqiang Pei

Lanzhou Institute of Chemical Physics

No.18 Tianshui Middle Rd., Lanzhou, 73000, China

Comments 1 :The authors have mentioned the properties of polyimide but have not mentioned the exact industrial applications. So please add a paragraph mentioning the applications of polyimides.

Response: Thanks for the suggestion. We have added the applications of polyimide in the third paragraph of the Introduction. [As a result, it is widely used in aerospace industry, triboelectric nanogenerators, and radiation-shielding materials. ]

Comments 2 :Everywhere, the authors have used the word tribological properties. I recommend the use of specific terms like wear rate and friction coefficient.

Response: We have translated some of the words in the article about tribological properties into specific terms. [In this case, three-body abrasive wear might be induced and the composites’ anti-wear performance are reduced.]

Reviewer 3 Report

Comments and Suggestions for Authors

The application of Metal-Organic Frameworks (MOFs) in the field of tribology has been paid increasing attention, which is actually still in the very initial stage especially for diamondoid MOFs. In the present study, the authors studied the effects of diamondoid MOFs incorporation on the tribological properties of polyimide and got very positive results regarding the modification role of MOFs. From the viewpoint of the reviewer, the manuscript can be accepted after revising the following minor issues:

1. The lay-out of figures in the Graphical Abstract should be reorganized to make them clearer.

2. Page 2 Line 43, “they been” should be “they have been”. And Line 44, just one citation of individual research [11] cannot reflect the interests in academic society. It is recommended to replace [11] with one review paper.

3. According to Fig.1, the particle size of prepared MOFs was between 700 nm and 1 μm, and the maximum content was 7wt.% in the present study. The reviewer is just wondering whether the addition of different content of MOFs has any influence on the imidization process.

4. In the COF evolution curves of Fig.3, Fig.5 and Fig.7, the maximum of Y axis can be decreased (e.g. to 0.6) to make the different curves more distinguishable.

5. According to Fig.5, the 3wt.% MOFs/PI composite still showed good wear resistance under 30 N, which illustrates the advantage of diamondoid MOFs as wear resistant modifiers. Further investigation is recommended to carry out regarding their synergy with traditional fillers.

Comments on the Quality of English Language

 Minor editing of English language required

Author Response

Dear Reviewer,

Thank you for your carefully review of our manuscript (polymers-2921735). We sincerely appreciate your very helpful comments and suggestions. The paper has been modified accordingly and all detailed changes in the paper are listed below point by point. We hope that you are satisfied with our responses to the comments. Thanks again for your efforts in processing our manuscript, and look forward to your reply soon!

With best regards,

Yours sincerely,

Xianqiang Pei

Lanzhou Institute of Chemical Physics

No.18 Tianshui Middle Rd., Lanzhou, 73000, China

Comments 1 :The lay-out of figures in the Graphical Abstract should be reorganized to make them clearer.

Response: The image has been redrawn.

Comments 2 :Page 2 Line 43, “they been” should be “they have been”. And Line 44, just one citation of individual research [11] cannot reflect the interests in academic society. It is recommended to replace [11] with one review paper.

Response: Thank you for your suggestion, we think your proposal is helpful. We have selected the review titled Recent advances in metal-organic frameworks for lubrication as reference.

Comments 3 : According to Fig.1, the particle size of prepared MOFs was between 700 nm and 1 μm, and the maximum content was 7wt.% in the present study. The reviewer is just wondering whether the addition of different content of MOFs has any influence on the imidization process.

Response: The addition of MOFs forms specific chemical bonds and hydrogen bonds with PAA, allowing it to grow on polyimide in-situ, and its content may have a certain effect on imimidation process under the influence of bonding phases.

Comments 4 : In the COF evolution curves of Fig.3, Fig.5 and Fig.7, the maximum of Y axis can be decreased (e.g. to 0.6) to make the different curves more distinguishable.

Response: Thanks for the suggestion, we have made changes to the picture

Comments 5: According to Fig.5, the 3wt.% MOFs/PI composite still showed good wear resistance under 30 N, which illustrates the advantage of diamondoid MOFs as wear resistant modifiers. Further investigation is recommended to carry out regarding their synergy with traditional fillers.

Response: You have great advice on what to do next, and we will explore the synergy between MOF and other traditional fillers in the next steps.